

# Radiomics machine learning based on asymmetrically prominent cortical and deep medullary veins combined with clinical features to predict prognosis in acute ischemic stroke: a retrospective study

Hongyi Li[1,2], Cancan Chang[3], Bo Zhou[2], Yu Lan[4], Peizhuo Zang[5], Shannan Chen[6], Shouliang Qi[6], Ronghui Ju[2] and Yang Duan[1,7]

[1] Dalian Medical University, Dalian, Liaoning, China
[2] Department of Radiology, The People's Hospital of China Medical University, The People's Hospital of Liaoning Province, Shenyang, Liaoning, China
[3] Department of Medical Imaging, Bozhou Hospital of Traditional Chinese Medicine, Bozhou, Anhui, China
[4] Department of Medical Imaging, Liaoning Cancer Hospital, Shenyang, Liaoning, China
[5] Department of Cerebrovascular Disease Treatment Center, The People's Hospital of China Medical University, The People's Hospital of Liaoning Province, Shenyang, Liaoning, China
[6] College of Medicine and Biological Information Engineering, Northeastern University, Shenyang, Liaoning, China
[7] Department of Radiology, Center for Neuroimaging, General Hospital of Northern Theater Command, Shenyang, Liaoning, China

Corresponding authors
Ronghui Ju, syyfsk1@163.com
Yang Duan, duanyang100@126.com

## ABSTRACT

**Background.** Acute ischemic stroke (AIS) has a poor prognosis and a high recurrence rate. Predicting the outcomes of AIS patients in the early stages of the disease is therefore important. The establishment of intracerebral collateral circulation significantly improves the survival of brain cells and the outcomes of AIS patients. However, no machine learning method has been applied to investigate the correlation between the dynamic evolution of intracerebral venous collateral circulation and AIS prognosis. Therefore, we employed a support vector machine (SVM) algorithm to analyze asymmetrically prominent cortical veins (APCVs) and deep medullary veins (DMVs) to establish a radiomic model for predicting the prognosis of AIS by combining clinical indicators.

**Methods.** The magnetic resonance imaging (MRI) data and clinical indicators of 150 AIS patients were retrospectively analyzed. Regions of interest corresponding to the DMVs and APCVs were delineated, and least absolute shrinkage and selection operator (LASSO) regression was used to select features extracted from these regions. An APCV-DMV radiomic model was created via the SVM algorithm, and independent clinical risk factors associated with AIS were combined with the radiomic model to generate a joint model. The SVM algorithm was selected because of its proven efficacy in handling high-dimensional radiomic data compared with alternative classifiers (*e.g.*, random forest) in pilot experiments.

**Results.** Nine radiomic features associated with AIS patient outcomes were ultimately selected. In the internal training test set, the AUCs of the clinical, DMV–APCV radiomic

and joint models were 0.816, 0.976 and 0.996, respectively. The DeLong test revealed that the predictive performance of the joint model was better than that of the individual models, with a test set AUC of 0.996, sensitivity of 0.905, and specificity of 1.000 ($P < 0.05$).

**Conclusions**. Using radiomic methods, we propose a novel joint predictive model that combines the imaging histologic features of the APCV and DMV with clinical indicators. This model quantitatively characterizes the morphological and functional attributes of venous collateral circulation, elucidating its important role in accurately evaluating the prognosis of patients with AIS and providing a noninvasive and highly accurate imaging tool for early prognostic prediction.

# INTRODUCTION

Acute ischemic stroke (AIS) is a disease in which occlusion or stenosis of the intracranial artery causes brain tissue ischemia and brain cell necrosis in the corresponding blood supply area. This disease occurs in people at many years of life, has a poor prognosis, is associated with high disability rates, and is currently recognized as the second leading cause of death worldwide (*Feigin et al., 2022*; *Jang et al., 2022*). Predicting patient outcomes by imaging examination is particularly important for developing accurate clinical treatment strategies and rehabilitation training regimens. The emergence of key imaging indicators such as the ischemic penumbra and the prediction of hemorrhagic transformation (*Leng et al., 2016*; *Wufuer et al., 2018*) has not only provided empirical support for the assessment of stroke conditions but also contributed to evaluations of the reperfusion and hemorrhagic transformation rates of brain tissue and improved the clinical outcomes and quality of life of patients (*Lee et al., 2021*).

The key to predicting the outcomes of AIS patients lies in developing accurate methods for assessing collateral circulation, among which imaging examinations are particularly important. Although digital subtraction angiography (DSA) can be used to directly image the brain vasculature, it is limited by its invasiveness (*Shaban et al., 2022*). Multiphase computed tomography angiography (mCTA) can be used to observe the vascular system in the brain through delayed multiphase scanning, but it is not effective in visualizing microvessels (*Menon et al., 2015*), and the use of delayed multiphase scanning requires larger doses of radiation (*Mnyusiwalla, Aviv & Symons, 2009*). Although magnetic resonance angiography (MRA) does not involve the use of radiation and has high sensitivity in detecting vessels at the proximal end of the circle of Willis—making it suitable for determining the degree and location of arterial stenosis—it has rarely been used to study collateral circulation (*Laviña, 2016*). Susceptibility weighted imaging (SWI) is currently the only imaging modality that can image intracerebral veins and accurately reflect changes in venous deoxyhemoglobin content. SWI has been widely used for intracranial

venous imaging. Previous studies have shown that deep medullary veins (DMVs) and asymmetrically prominent cortical veins (APCVs) are among the imaging indicators that can represent collateral circulation (*Chen et al., 2020*; *Huang et al., 2022*) and can be clearly displayed on SWI sequences.

The APCV and DMV appear dilated on SWI *via* an increase in magnetic susceptibility due to an increase in the deoxyhemoglobin content in the blood vessels, resulting in the development of obvious low signals in veins. These two sets of veins also maintain a stable intracerebral blood volume and are helpful for assessing the prognosis of AIS patients. Identification and assessment of DMVs have also been applied to investigations of cognitive impairment (*Chen et al., 2023*), white matter hyperintensity (*Keith et al., 2017*), cerebral small vessel disease (*Han, Huang & Shan, 2014*; *Liao et al., 2024*), and Sturge Weber syndrome (*Jeong et al., 2024*). Among them, cerebral venous collateral circulation significantly influences neurological recovery in stroke patients by maintaining hemodynamic stability in the ischemic semidark zone region (*Xiang et al., 2023*; *Kao, Tsai & Hasso, 2012*). Disruption of DMV continuity is strongly associated with cognitive impairment in cerebral small vessel disease (CSVD), further supporting the pathophysiological role of the venous system in ischemic events (*Chen et al., 2023*; *Liao et al., 2024*). Previous studies have focused on manual assessments of DMV (*Li et al., 2023*) and APCV features, which are time consuming and can yield subjective results. Therefore, it is particularly urgent to find more objective and accurate methods for evaluating the characteristics of cerebral vein collaterals. Rapid developments in radiomics technology in recent years have allowed the accurate extraction of radiomic features from complex medical images through quantitative analyses of big medical imaging data (*Mayerhoefer et al., 2020*), achieving accurate diagnoses, treatment response evaluations, and predictions of disease prognosis.

Numerous studies have investigated the application of deep learning (DL) methods and machine learning (ML) methods in radiomics for AIS from diverse perspectives. Specifically, three-dimensional (3D) convolutional neural network (CNN) algorithms have demonstrated efficacy in AIS lesion segmentation and etiological subtype classification, achieving superior accuracy (*Kim et al., 2023*; *Kousar et al., 2025*; *Bal et al., 2024*). Radiomic features associated with the ischemic penumbra, such as the R score, exhibit predictive value for thrombolytic efficacy in AIS patients (*Tang et al., 2020*); moreover, quantitative radiomic metrics enable the precise delineation of penumbral boundaries (*Zhang et al., 2020*). Furthermore, a radiomic model based on the sparse representation method (SRM) has shown promising potential in predicting postinfarction volume expansion trends (*Wu et al., 2024*). Delta radiomic models have demonstrated efficacy in predicting hemorrhagic transformation following intravenous thrombolysis for acute cerebral infarction (*Wu et al., 2025*). Collectively, these studies highlight the diagnostic and therapeutic potential of radiomics in AIS. However, existing studies exhibit a critical knowledge gap: no studies have systematically investigated the correlation between the dynamic evolution of intracerebral venous collateral circulation and AIS prognosis. Our methodological prioritization of ML over DL methods, including state-of-the-art (SOTA) and 3D-CNN algorithms, stems from three inherent limitations of DL: (1) the black-box nature of DL impedes interpretability

of extracted radiomic features, failing to meet clinical demands for model transparency; (2) DL requires large-scale annotated datasets for training, which renders it unsuitable for small-sample studies ($n < 200$ in our study); (3) the computational costs of model training are prohibitive compared with the efficient implementation of ML on standard hardware.

This study employs radiomics-based machine learning methods to investigate the associations between cerebral venous collateral circulation and clinical outcomes in patients with AIS. We propose a novel joint predictive model that combines radiomic features of APCV and DMV with clinical indicators. This model quantitatively characterizes the morphological and functional attributes of venous collateral circulation, thereby overcoming the limitations of conventional subjective scoring systems. Furthermore, we validated cerebral venous collateral circulation as an independent prognostic biomarker, establishing an imaging foundation for personalized therapeutic strategies. We anticipate that this predictive model will enable clinicians to assess patient prognosis through a single MRI scan, facilitating the development of individualized long-term treatment plans. By optimizing medication timing and therapeutic interventions, this approach has the potential to increase overall treatment efficacy and quality of life in AIS patients.

## MATERIALS & METHODS

### Participants

The clinical and imaging data of patients who were diagnosed with AIS and hospitalized in the Department of Neurology of Liaoning Provincial People's Hospital between January 2018 and June 2023 were retrospectively analyzed. The functional outcomes of the patients were evaluated with the modified Rankin scale (mRS). The study adhered to Helsinki principles for studies involving human subjects and received approval from the ethical committee of the People's Hospital of Liaoning Province (approval no. (2024) H004). Since this was a retrospective study, additional informed consent was waived.

The inclusion criteria were as follows: (I) the diagnostic criteria of the Chinese guidelines for the diagnosis and treatment of acute ischemic stroke 2023 (*Chinese Society of Neurology et al., 2024*); and (II) head MR imaging data obtained within 72 h after onset, including T1-weighted imaging (T1WI), T2-weighted imaging (T2WI), T2 fluid attenuated inversion recovery (FLAIR) imaging, MRA, DWI, and SWI.

The exclusion criteria were as follows: (I) other abnormal brain MRI results, such as cerebral hemorrhage, brain space-occupying lesions, head trauma, and cerebrovascular malformations; (II) infarction combined with hemorrhage; (III) MR images with severe artifacts that would affect the results of the image analysis; (IV) previous stroke with sequelae and a basic mRS score $\geq 1$; and (V) loss to follow-up.

The patient screening and research workflows are shown in Figs. 1 and 2. To balance the data during model training, we ensured a 1:1 ratio between groups. A total of 150 AIS patients (109 males and 41 females; aged 65.13 $\pm$ 10.99 years) were included in this study, and their outcomes were grouped according to the mRS score. Patients with a good outcome were defined as those with an mRS score ranging from 0–2 points, and those with a poor outcome were defined as those with an mRS score $\geq 3$ points. Thus, a total of 75
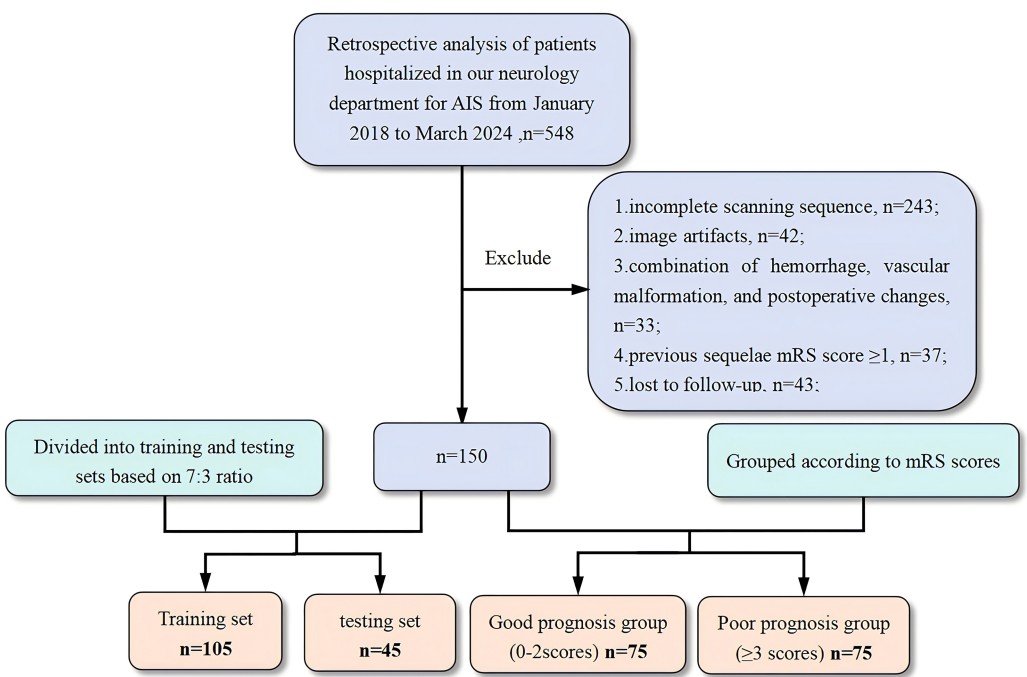

**Figure 1  Flowchart of the patient selection process.** AIS, acute ischemic stroke; mRS, modified Rankin scale.

patients were included in the poor outcome group and the good outcome group. In the radiomic study, to verify the performance of the machine learning model, the patient data were randomly allocated to the training and test sets at a 7:3 ratio ($n = 105$ and $n = 45$, respectively).

## Image data acquisition

All patients were scanned with a Discovery 750 3.0T MRI scanner equipped with an 8-channel coil (GE HealthCare, Chicago, IL, USA). The patient was placed in the supine position with the head facing forward and scanned parallel to the anterior commissure-posterior commissure line from the foramen magnum to the cranial vault. The scan sequences included conventional T1WI, T2WI, FLAIR, DWI and SWI sequences. The SW images were subjected to minimum intensity projection to obtain the mIP map. The scan parameters of each sequence are shown in Table S1. All scan data were uploaded to an ADW 4.7 workstation (GE HealthCare, Chicago, IL, USA) for postprocessing and analysis.

## Clinical data collection

Data were collected as previously described in *Li et al. (2023)*. Specifically age at admission, sex, systolic and diastolic blood pressure, fasting glucose, cholesterol, total homocysteine (tHcy), National Institutes of Health Stroke Scale (NIHSS) score at admission, and mRS score three months post discharge. Imaging indicators included the DMV score, DMV symmetry, cerebral cortical vein quantification score, infarct area, and white matter hyperintensity (WMH) score.

Peer**J**

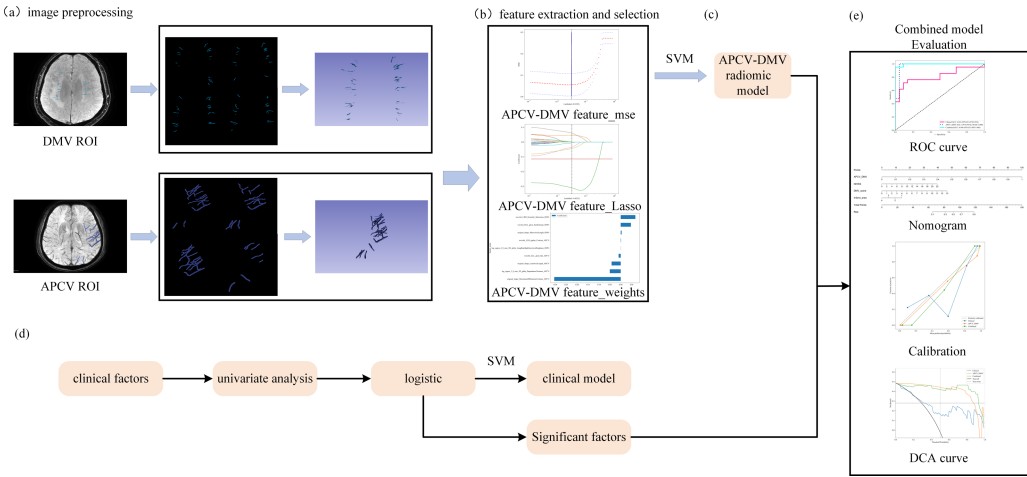

**Figure 2** **Study flowchart.** (A) 3D Slicer software was used to outline regions of interest corresponding to the DMVs on SWI and regions of interest corresponding to the APCVs on SWI mIP images. (B, C) LASSO feature screening for radiomic model construction *via* the SVM machine learning algorithms. (D) Clinical model construction process. (E) Combined model construction and performance curves. DMV, deep medullary vein; SWI, susceptibility-weighted imaging; APCV, asymmetrically prominent cortical vein; LASSO, least absolute shrinkage and selection operator; SVM, support vector machine.

(1) DMV score: The DMVs that drain the frontal lobe and anterior parietal lobe white matter are distributed in a vertical fan shape next to the anterior horn of the lateral ventricle. The DMVs that drain the posterior parietal lobe and the occipital lobe white matter are distributed throughout the posterior horn or triagonal area of the lateral ventricle, primarily in the upper and lower planes relative to the lateral paraventricular plane. Therefore, we divided the planes above and below the lateral paraventricular plane into six regions (Fig. 3). The DMV score was calculated as the score for each area, determined as follows (*Li et al., 2023*; *Wang et al., 2024*; *Zhang et al., 2017*): 0 points = each vein was continuous, and the signal was uniform; 1 point = each vein was continuous, but one or more veins had uneven signals; 2 points = one or more veins were discontinuous, manifesting as a punctate low signal; 3 points = none of the veins were continuous.

(2) Determination of DMV symmetry: DMVs were considered symmetrical if they were observed in the same area near the lateral ventricles with similar degrees of dilation and differed between the hemispheres by no more than five veins (*Li et al., 2023*).

(3) White matter hyperintensity (WMH) evaluation: On FLAIR imaging, WMH evaluation was performed according to the Fazekas scale (0–6 points), as previously described (*Wang et al., 2022*).

(4) Infarct area: The area was measured and classified on the slice showing the maximum infarct area on DWI: (I) lacunar infarction: $\leq 1.5$ cm$^2$; (II) small infarction: 1.6–3.0 cm$^2$; (III) moderate infarction: 3.1–5.0 cm$^2$ and occupying less than one lobe of the brain; (IV) large-area infarction: occupation of more than one lobe of the brain or an infarct area larger than 5.0 cm$^2$.

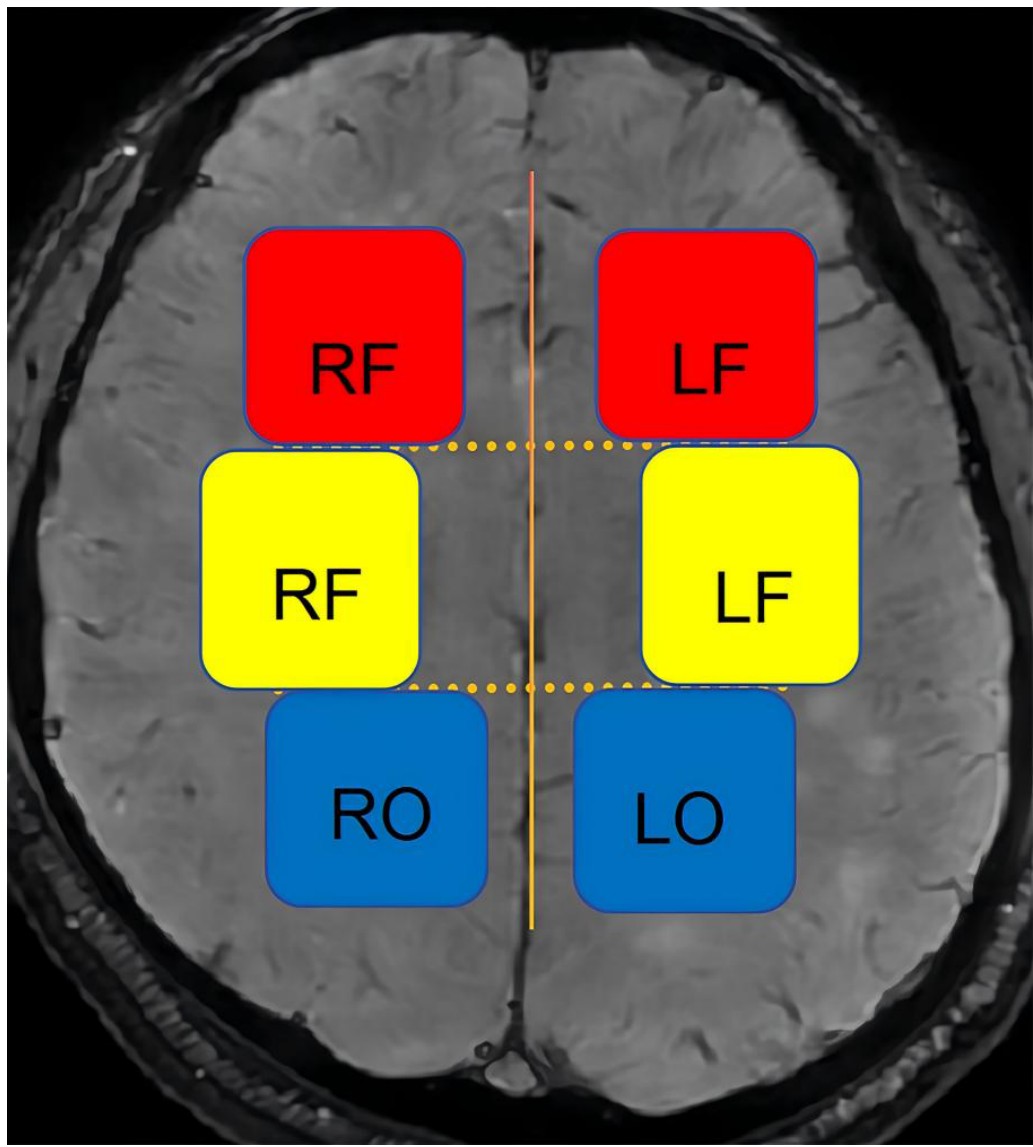

**Figure 3  Schematic diagram of the DMV scoring partition.** LF, left frontal; RF, right frontal; LP, left parietal; RP, right parietal; LO, left occipital; RO, right occipital.

(5) Quantitative cerebral cortical vein score: The prominent cortical veins were scored on the SWI mIP images with reference to the Alberta Stroke Project Early CT Score (ASPECTS). Specifically, the brain is divided into the following areas: L = lenticular nucleus; M1 = anterior middle cerebral artery (MCA) cortex; M2 = MCA cortex on the lateral side of the insula; M3 = cortical area at the posterior part of the MCA; M4–M6: cortical areas above the respective M1–M3 areas. Each area is given a score of 1 point, with a maximum prominent vessel sign (PVS) score of 7 points. The value of normal cortical veins is 0, whereas a score of seven indicates diffuse dilatation of the cortical veins. Figure 4 shows a graph of asymmetrically prominent cortical veins.

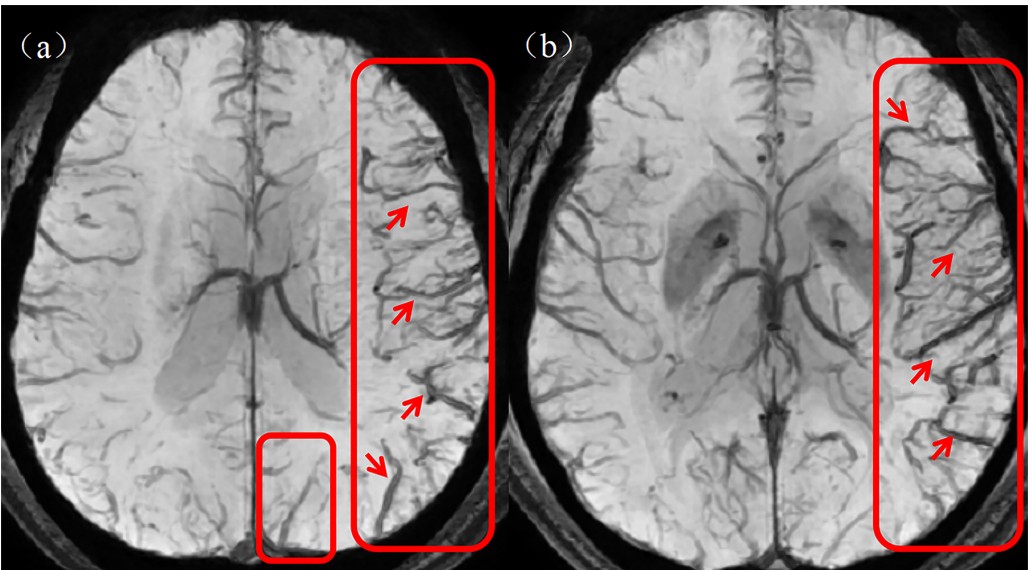

**Figure 4** **Asymmetrically prominent cortical veins.** (A, B) The rectangular boxes and short arrows in the figure show asymmetrically prominent cortical veins on SWI mIP images. SWI, susceptibility weighted imaging; mIP, minimum intensity projection.

## Region of interest segmentation

The region of interest (ROI) was manually segmented slice by slice by two junior radiologists with 3 and 5 years of experience in 3D Slicer software (version 5.2.2, https://www.slicer.org/). The segmentation results were reviewed and validated by a senior radiologist with 15 years of experience. On each consecutive imaging slice from the SWI sequence, the ROI was delineated along the contour of each DMV to cover the entire volume of the vein. Up to five DMV ROIs annotated with their morphological characteristics were delineated in each of the six regions according to the DMV score, as shown in Figs. 5A–5E. On each consecutive imaging slice of the SWI mIP images, each ROI was delineated according to the profile of the APCV to cover the entire volume of the vein (Figs. 5F–5J). These ROIs were subsequently used for radiomic feature extraction and screening.

## Data preprocessing

The data were randomly allocated at a ratio of 7:3 to a training set and a test set. All the data in the training set were used to train the predictive model, whereas those in the test set were used to independently evaluate the performance of the model. Owing to scanner or acquisition differences, unlike other schemes, spatial normalization is typically used to reduce the effect of differences in voxel spacing. To solve this problem, we used a fixed-resolution resampling method in our experiments, resampling all the images to a size of $1 \times 1 \times 1$ mm, $Z$ score normalization, Laplacian–Gaussian filtering ($\sigma = 2$–5) and wavelet transform to extract multiscale features.

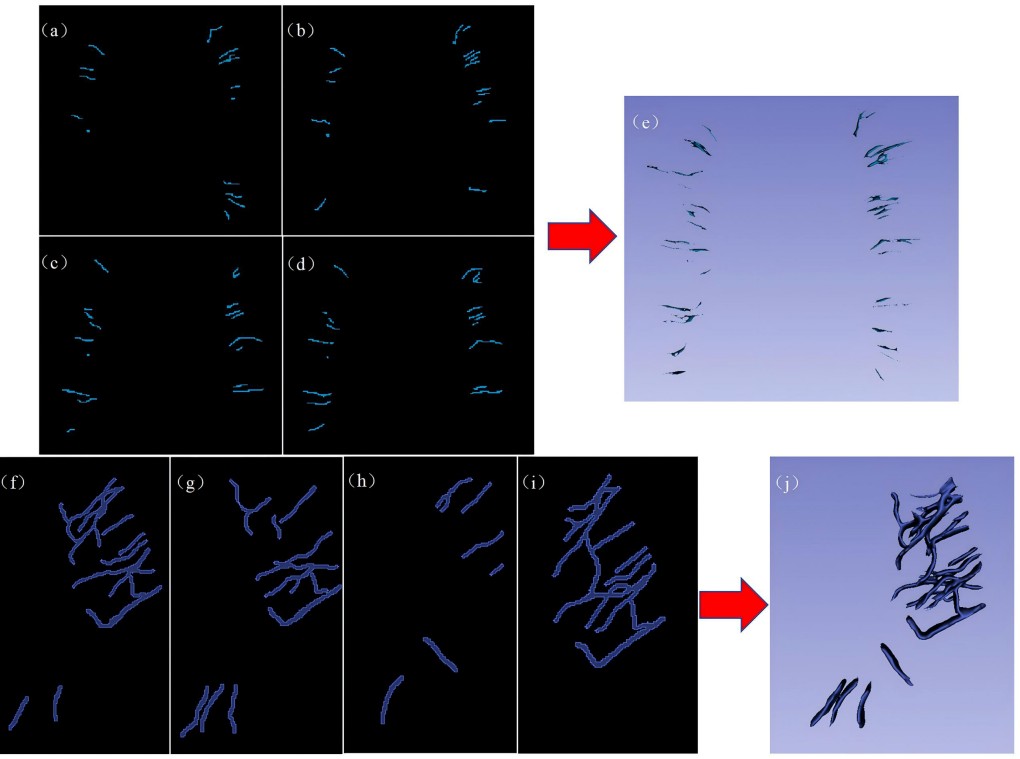

**Figure 5    Representative examples of DMV labeling and APCV labeling.** (A–D) 2D image of the regions of interest of the DMVs delineated in 3D Slicer software, and (E) 3D stereo image of the entire DMV region of interest. (F–I) 2D images of the regions of interest of the APCVs delineated in 3D Slicer software. (J) 3D stereo image of the entire APCV region of interest. DMV, deep medullary vein. APCV, asymmetrically prominent cortical vein.

## Feature extraction

Radiomic features were extracted from each ROI *via* the PyRadiomics package (version 3.1.0, https://pyradiomics.readthedocs.io) in Python (version 3.7.0). The MR images and corresponding ROIs were simultaneously used to extract radiomic features, which could be divided into three groups: (I) geometric features; (II) intensity features; and (III) texture feature. To address the scale heterogeneity among the artificial radiometric features, the features were normalized *via* the $Z$ score method. To obtain high-throughput features, nonlinear intensity filtering was performed at the voxel level with the Laplacian−Gaussian filter (with sigma values set to 2, 3, 4, and 5) for first-order statistics and eight wavelet transform algorithms (LLL, LLH, LHL, LHH, HLL, HLH, HHL, and HHH) for texture features.

## Feature selection

First, the Mann–Whitney $U$ test was used to identify radiomic features in the training set that were significantly different between the groups ($P < 0.05$). Next, Person's rank correlation analysis was used to calculate the correlation coefficients between the features, and one feature from a pair of features with a correlation coefficient greater than 0.9
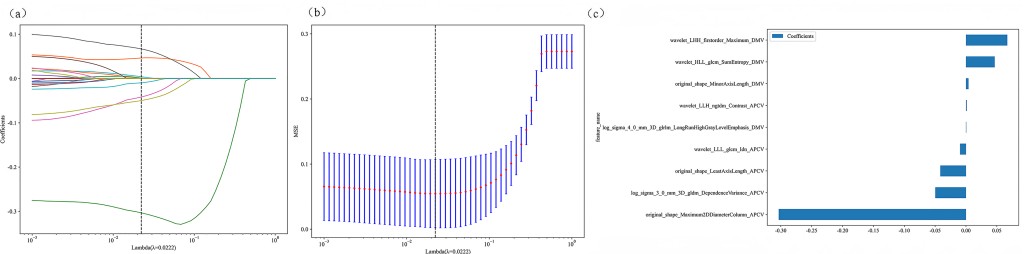

**Figure 6 Radiomic model 10-fold cross-validation and non-zero coefficient characterization plots.**
(A–B) Coefficients and MSEs derived from 10-fold cross-validation. (C) Features with non-zero coefficient selected *via* least absolute shrinkage and selection operator regression.

was retained. The minimum redundancy–maximum relevance (mRMR) algorithm was used to retain features with high correlations and low redundancy. Finally, the least absolute shrinkage and selection operator (LASSO) algorithm was used for further feature screening. The optimal λ value was determined with 10-fold cross-validation and the minimum standard, and the features with nonzero coefficients were retained.

LASSO regression selects features by minimizing the following loss function:

$$\min_{\beta} \left( \frac{1}{2N} \sum_{i=1}^{N} (y_i - \beta^T X_i)^2 + \lambda \parallel \beta \parallel_1 \right)$$

where $N$ is the number of samples, λ is the regularization parameter, and the optimal value is determined *via* 10-fold cross-validation (Fig. 6).

## Clinical factor analysis

To select clinical factors significantly associated with patient outcomes, univariable analysis was conducted on 13 clinical factors to identify those with significant differences between the good outcome group and the poor outcome group, after which multivariable regression analysis was performed to identify independent risk factors for poor outcomes in AIS patients.

## Model construction
### *Radiomic model construction and validation*

The Python scikit-learn package was used for LASSO regression modeling. After LASSO feature screening, we input the final features into a support vector machine (SVM) algorithm (*Liu et al., 2023*) to construct the risk regression model. Here, we used 5-fold cross-validation to obtain the final APCV-DMV radiomic model. An SVM was chosen because of its excellent classification performance for high-dimensional small-sample data and its superior generalization performance. In addition, we compared different feature selection and model-building algorithms. Descriptions and results are shown in Table S2.

Using the SVM kernel function:

$$K(x_i, x_j) = \exp(-\gamma \parallel x_i \parallel - x_j^2)$$

where $\gamma$ is optimized *via* grid search (range: $10^{-3}$ to $10^3$).

*Clinical model construction and validation*

To maintain algorithmic consistency across models, the same SVM algorithm was selected for the clinical models. Therefore, after univariable and multivariable regression analyses of the clinical factors, the factors with $P$ values $< 0.05$ were input into the SVM machine learning algorithm with 5-fold cross-validation to construct the clinical model.

*Combined model*

To visually and effectively evaluate the incremental prognostic value of the APCV-DMV radiomic model over the clinical risk factor model, a joint model was proposed and assessed in the test set. This joint model integrated the APCV-DMV radiomic model and clinical risk factors determined from logistic regression analysis.

## Statistical analysis

All the data were analyzed with the Python statamodels package (version 0.13.2), and a $P$ value $< 0.05$ was considered to indicate statistical significance. For continuous variables, we used the $t$ test or the Mann–Whitney $U$ test to analyze differences among groups; for categorical variables, we used the chi-square test or Fisher's test for between-group comparisons. For screening baseline clinical characteristics, univariable analysis was performed first, followed by multivariable regression analysis. For radiomic characteristics, we used the Mann–Whitney $U$ test and Pearson's rank correlation analysis, the mRMR algorithm, the LASSO algorithm, and the cross-validation method for screening. The screened features were input into the SVM algorithm to construct a risk model. The DeLong test was used to compare performance metrics between the models. Moreover, we generated receiver operating characteristic (ROC) curves to evaluate the prediction model. The area under the curve (AUC), sensitivity (Sen), specificity (Spe), accuracy (ACC), confidence interval (CI) and precision were also calculated and analyzed.

## RESULTS

### Clinical characteristics

Following the application of the inclusion and exclusion criteria, a total of 150 patients were included in the study, with a mean age of $65.13 \pm 10.99$ years. There were 109 males and 41 females, including 75 patients in the good outcome group and 75 patients in the poor outcome group. The baseline clinical data of the patients in the two groups and the results of their comparisons are shown in Table 1. Table 2 lists the results of multivariable regression analysis of the clinical characteristics, NIHSS score $P = 0.000$, DMV score $P = 0.004$, infarct size $P = 0.041$, prominent cortical vein score $P = 0.747$, and DMV symmetry $P = 0.394$ in the training set.

### Radiomics model

Radiomic features were extracted from the DMV- and APCV-delineated ROIs. A total of 1,197 features were extracted, including 234 first-order features, 14 shape features and 949 texture features. A total of 431 features were screened with the $t$ test or the Mann–Whitney $U$ test and Pearson's correlation test. The use of the mRMR algorithm resulted in the retention of 20 features with high correlation and low redundancy. Finally,

**Table 1 Baseline clinical characteristics of the patients in our cohort.**

| Characteristics | Outcome grouping | | | P value |
| --- | --- | --- | --- | --- |
| | Overall, $n = 150$ | Good group, $n = 75$ | Poor group, $n = 75$ | |
| Age | 65.13 ± 10.99 | 63.62 ± 11.39 | 66.69 ± 10.40 | 0.087 |
| Sex | | | | 0.92 |
| Male | 109(72.67) | 55(73.33) | 57(72.00) | |
| Female | 41(27.33) | 20(26.67) | 21(28.00) | |
| Systolic blood pressure | 156.98 ± 23.45 | 156.34 ± 24.78 | 157.64 ± 22.14 | 0.737 |
| Diastolic blood pressure | 88.54 ± 17.16 | 88.11 ± 16.01 | 88.99 ± 18.36 | 0.618 |
| Fasting glucose | 7.27 ± 3.59 | 6.91 ± 3.39 | 7.65 ± 3.77 | 0.062 |
| Cholesterol | 4.61 ± 1.15 | 4.69 ± 1.15 | 4.53 ± 1.15 | 0.382 |
| Total homocysteine level | 20.94 ± 12.05 | 22.32 ± 13.91 | 19.53 ± 9.68 | 0.297 |
| NIHSS score | 5.02 ± 4.73 | 2.91 ± 2.96 | 7.19 ± 5.22 | <0.001 |
| Infarct area | | | | <0.001 |
| Lacunar | 41(27.33) | 32(42.67) | 10(13.33) | |
| Small | 34(22.67) | 21(28.00) | 13(17.33) | |
| Moderate | 17(11.33) | 7(9.33) | 9(12.00) | |
| Large | 58(38.67) | 15(20.00) | 43(57.34) | |
| Prominent cortical vein score | 2.59 ± 1.79 | 1.95 ± 1.37 | 3.26 ± 1.93 | <0.001 |
| DMV score | 5.91 ± 3.47 | 4.58 ± 3.03 | 7.27 ± 3.38 | <0.001 |
| WMH score | 3.39 ± 1.75 | 3.11 ± 1.69 | 3.68 ± 1.78 | 0.053 |
| DMV symmetry | | | | <0.001 |
| Symmetrical | 57(38.00) | 39(52.00) | 18(24.00) | |
| Asymmetrical | 93(62.00) | 36(48.00) | 57(76.00) | |

Notes.
$P < 0.05$ is considered to indicate a statistically significant difference, and the data are presented as the means ± standard deviations or $n$ (%).
NIHSS, National Institutes of Health Stroke Scale; DMV, deep medullary vein; WMH, white matter hyperintensity.

**Table 2 Multivariable logistic regression analysis of clinical factors in the training set.**

| Variable | OR (95%CI) | P value |
| --- | --- | --- |
| NIHSS | 1.036 (1.02–1.052) | 0.000 |
| DMV score | 1.044 (1.019–1.069) | 0.004 |
| Prominent cortical veins score | 1.01 (0.961–1.061) | 0.747 |
| Infarct area | 1.081 (1.015–1.15) | 0.041 |
| DMV symmetry | 1.089 (0.923–1.285) | 0.394 |

Notes.
$P < 0.05$ is considered to indicate a statistically significant difference.
NIHSS, National Institutes of Health Stroke Scale; DMV, deep medial vein; OR, odds ratio; CI, confidence interval.

through LASSO regression, tenfold cross-validation was performed using the minimum criterion to determine the optimal λ value (Figs. 6A and 6B). Ultimately, nine DMV and APCV radiomic features with nonzero coefficients were selected (Fig. 6C). Finally, the selected features were input into the SVM algorithm, and the radiomic feature model was constructed through fivefold cross-validation. The ROC curves of the model in the

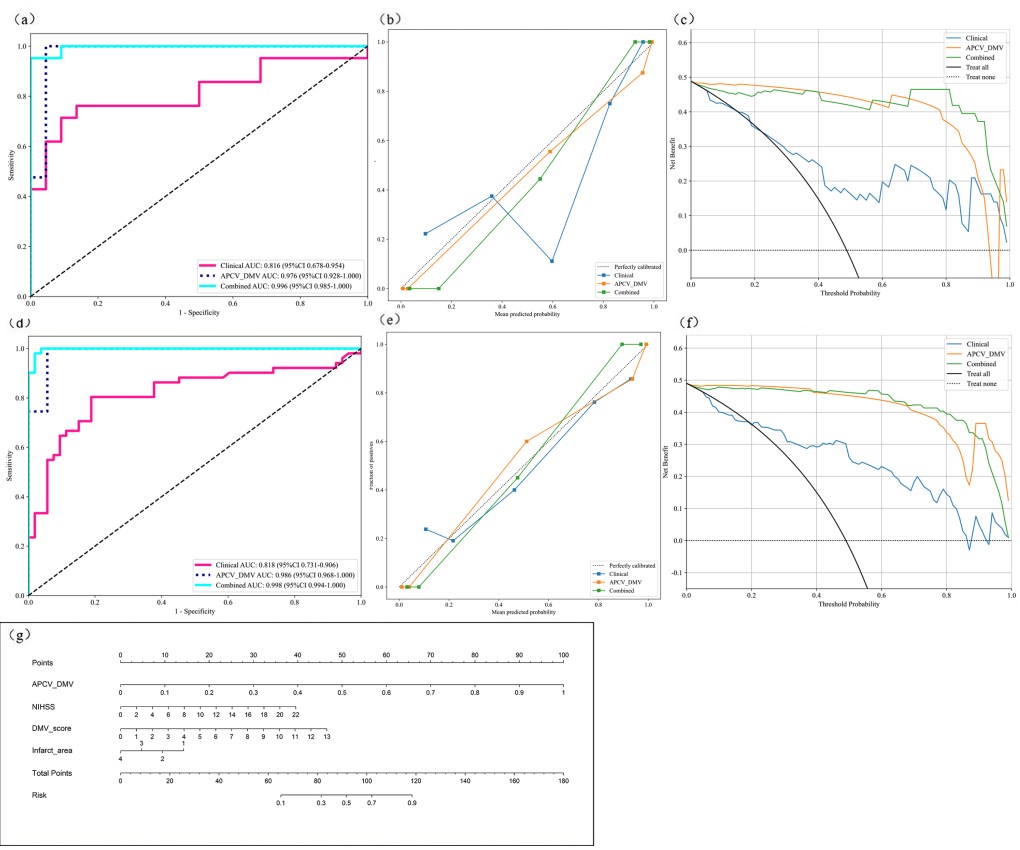

**Figure 7  Model performance analysis.** (A) ROC curves of the three prediction models in the test set; (B) model calibration curves in the test set; (C) model decision curves in the test set; (D) ROC curves of the three prediction models in the training set (E) model calibration curves in the training set; (G) prognostic nomogram developed from the combined model for predicting the outcomes of AIS patients. ROC, Region of interest. AIS, acute ischemic stroke.

training set (AUC =0.986) and the test set (AUC = 0.976) are shown in Figs. 7A–7D. Table 3 shows that in the test set, Sen = 0.952, Spe = 0.955, PPV = 0.952, and NPV = 0.955. In addition to the AUC, sensitivity, and specificity, F1 scores were calculated to assess model balance comprehensively. Fivefold cross-validation revealed a mean AUC of 0.992 (standard deviation ± 0.006) for the combined model.

## Clinical model

Multivariate regression analysis of the clinical characteristics that differed between the groups revealed that the NIHSS score, DMV score and infarct area were independent risk factors for poor AIS patient outcomes. Therefore, these three factors were selected as inputs into the SVM algorithm to construct the clinical model. The ROC curves of the model in the training set (AUC = 0.818) and test set (AUC = 0.816) are shown in Figs. 7A and 7D. In the test set, Sen=0.714, Spe=0.864, PPV=0.833, and NPV=0.76, as shown in Table 3.

## Combined model

By integrating the radiomics model and the clinical model, a joint model was established. Table 3 shows the performance indicators of the joint model, with AUC = 0.998 in the training set and AUC = 0.996 in the test set. The ROC curves are shown in Figs. 7A and 7D. Moreover, in the test set, Sen = 0.905, Spe = 1.000, PPV = 1.000, and NPV = 0.917. Figure 7G shows the nomogram of the joint model incorporating clinical features and APCV-DMV radiomic features. Here, the scores of the individual elements are added to obtain the total score. The predicted risk values shown correspond to the probabilities of a poor outcome for AIS patients. Figure 7B and e shows the calibration curves for the model in the training and test sets. The plots show that in both sets, the consistency between the predicted and observed outcomes of AIS patients was consistent.

The individual and joint models were also evaluated *via* decision curve analysis (DCA). As shown in Figs. 7C and 7F, DCA of the clinical model, the APCV-DMV radiomic model and the joint model revealed that the joint model had the greatest net clinical benefits in predicting the outcomes of AIS patients before treatment. Similarly, the DeLong test suggested that the joint model could best predict the outcomes of AIS patients ($P < 0.05$). The AUC value for predicting the outcomes of AIS patients was significantly greater with the joint model than with the other two models in both the training set and the test set (Table S3).

## DISCUSSION

Our study revealed that the combined clinical feature–radiomics model had the best prediction performance for the outcomes of AIS patients, demonstrating better sensitivity and specificity than the individual models did. Using the SVM algorithm, the vascular features of the cerebral cortical veins and deep medullary veins were extracted (*e.g.*, first-order features, shape features and texture features), thus reflecting the information at the collateral circulation level of the cerebral veins and allowing a more objective and accurate prediction of the outcomes of AIS patients from the level of the vasculature. The results revealed that wavelet-LHH-first-order-maximum and wavelet-LHH-glcm-sum entropy features of the DMV, as well as the APCV original-shape-Maximum2DDiameterColumn and 3D-fldm-DependenceVariance features, contributed the most to the model, suggesting that the microheterogeneity of the venous collateral branches is critical for prognosis. The values of the clinical model, the APCV-DMV radiomic model and the combined model in predicting AIS patient prognoses were visually displayed through the ROC curve. The combination of clinical features and imaging features of the cerebral cortical veins and deep medullary veins led to greater AUC values, indicating that clinical features play a supplementary role in predicting patient outcomes. During the development of AIS, the venous return of the cerebral cortical veins and deep medullary veins from the superficial and deep veins, respectively, reflects changes in the venous circulatory system, whereas the internal cerebral venous system, as a part of collateral circulation, is involved in maintaining blood volume balance. These results confirm the value of the APCVs and DMVs in predicting the outcomes of AIS patients, providing an important basis for their use as prognostic imaging markers of the disease.

**Table 3** Comparison of the performance of the clinical model, APCV-DMV radiomic model and combined model for predicting the outcomes of AIS patients.

| Model | Training set ($n = 105$) | | | | | | | Test set ($n = 45$) | | | | | | |
|---|---|---|---|---|---|---|---|---|---|---|---|---|---|---|
| | AUC (95%CI) | Acc | Sen | Spe | PPV | NPV | F1 | AUC (95%CI) | Acc | Sen | Spe | PPV | NPV | F1 |
| Clinical | 0.818 (0.7311–0.9056) | 0.769 | 0.725 | 0.811 | 0.787 | 0.754 | 0.755 | 0.816 (0.6778–0.9543) | 0.791 | 0.714 | 0.864 | 0.833 | 0.760 | 0.769 |
| APCV-DMV | 0.986 (0.9682–1.0000) | 0.962 | 0.980 | 0.943 | 0.943 | 0.980 | 0.962 | 0.976 (0.9285–1.0000) | 0.953 | 0.952 | 0.955 | 0.952 | 0.955 | 0.952 |
| Combined | 0.998 (0.9936–1.0000) | 0.971 | 0.980 | 0.962 | 0.962 | 0.981 | 0.971 | 0.996 (0.9854–1.0000) | 0.953 | 0.905 | 1.000 | 1.000 | 0.917 | 0.950 |

**Notes.**

AIS, acute ischemic stroke; DMV, deep medial vein; APCV, asymmetrically prominent cortical vein; AUC, area under the curve; Sen, sensitivity; Spe, specificity; Acc, Accuracy; PPV, positive predictive value; NPV, negative predictive value.

The poor outcome group had a greater mean APCV score and a more asymmetrical distribution of DMVs than the good outcome group did, indicating that the number and diameter of cerebral cortical veins and deep medullary veins were greater in the poor outcome group. Venous dilatation indicates an increase in the level of deoxyhemoglobin in the blood vessels. An increased cellular oxygen uptake rate indicates poor perfusion of the infarcted area, under which the blood volume cannot be stabilized and the oxygen supply demand of brain tissue cannot be met (*Sun et al., 2014*), thus reducing the probability of neuronal survival. Therefore, good collateral circulation is the core element in the treatment and evaluation of stroke patients. Its manifestations are not limited to the arterial system but can also be found in the venous system and microcirculation in the brain (*Horie et al., 2011*; *Mucke et al., 2015*). Therefore, the status of cerebral cortical veins and deep medullary veins can be used as indicators of hypoperfusion status in stroke patients, which also reflects insufficient compensatory functioning of the collateral circulation (*Chen et al., 2015*; *Jing et al., 2021*), which is consistent with our study results.

Studies have confirmed that there is a close association between APCVs and poor outcomes in AIS patients (*Lu et al., 2021*; *Wang et al., 2021*). Furthermore, the presence of APCVs often predicts poor blood perfusion in the ischemic penumbra area, which could be a harbinger of early deterioration of neurological function (*Huang et al., 2022*; *Kao, Tsai & Hasso, 2012*; *Li et al., 2020*). Previous studies have shown that patients with severe large vessel stenosis or occlusion have a greater probability of developing extensive APCVs (*Jiang et al., 2021*; *Xiang et al., 2023*). In our study, the ASPECTS scoring system was used to quantitatively score dilated cerebral cortical veins, and the results revealed that the score in the poor outcome group was greater than that in the good outcome group, indicating that the number of dilated cortical veins and the extent of dilatation are greater in patients with poor outcomes, which is consistent with the results of previous studies.

The degree of continuity or injury of DMVs is related to the pathogenesis and total burden of CSVD, the number of cerebral microbleeds and the number of perivascular spaces in the basal ganglia (*Wang et al., 2024*). Our previous study confirmed that visual assessment of DMVs *via* SWI was helpful for predicting the outcomes of AIS patients. Therefore, in this study, we incorporated DMV-related imaging indicators into a clinical prediction model. *Han, Huang & Shan (2014)* noted that an asymmetric distribution of DMVs or the brush sign was associated with poor outcomes in AIS patients, which is consistent with the results of our univariable analysis. In the poor outcome group, the DMVs were mostly asymmetrically distributed; however, after multivariable regression analysis, we found that DMV asymmetry was not an independent risk factor for poor AIS prognosis, whereas the DMV visualization score was, which further confirmed the association between DMVs and poor AIS patient outcomes.

Interruption of DMV continuity and impaired venous drainage reduce the oxygen and nutrient supply to affected brain tissues, which in turn promotes worsening of ischemic injury (*Wang et al., 2024*; *Xu et al., 2019*). From a pathological perspective, our analysis suggests that collagen deposition in the wall of venules, increased venous pressure, and wall edema all lead to wall stenosis, which provides a mechanistic explanation for the decreased visibility or discontinuity of the DMVs on SWI (*Brown et al., 2002*; *Keith et al., 2017*;

*Moody et al., 1997*). In addition, during cerebral tissue ischemia, cell metabolism disorders trigger inflammation, resulting in increased intracerebral venous pressure. These changes further aggravate stenoses of the venous lumen and reduce the continuity of the DMVs on SWI. According to the above analyses, we can conclude that the relationships among DMV injury, intermittent DMV development and ischemic injury are mutually beneficial.

However, both the APCV assessment and the DMV visualization score are obtained manually and are therefore limited by a certain degree of subjectivity. Our study applied cutting-edge radiomic methods to more objectively discover many features that cannot be identified by the human eye and explored a one-stop prognostic prediction model based on the internal cerebral venous circulatory system and clinical data in AIS patients. This model could provide clinicians with prognostic information in the early stages of the disease, better meet daily clinical needs and offer a convenient tool for clinical application.

This study is limited in that the sample size of the single-center study still needs to be supplemented, and multicenter validation and expansion of the sample size will be conducted in the future. Manual outlining of ROIs may introduce bias, and automatic segmentation methods to extract and identify deep medullary veins and cerebral cortical veins should be considered in the future to shorten the prediction time of the model before the physician makes a decision. Although the interpretability of the model is better than that of deep learning, the biological significance of some texture features remains to be explored.

In summary, our study revealed that the APCVs and DMVs can be used as imaging markers for predicting the outcomes of AIS patients. Notably, a joint prediction model established on the basis of radiomic features of the APCVs and DMVs and clinical patient characteristics can effectively predict the 3-month outcomes of AIS patients in the early stages of the disease. These findings may serve as a powerful basis for clinical decision-making.

## CONCLUSIONS

In summary, our study revealed that APCVs and DMVs can be used as imaging markers for predicting the outcomes of AIS patients, and for the first time, the radiomic features of APCVs/DMVs were integrated to reveal the independent predictive value of venous side branches in AIS prognosis. The AUC of the combined prediction model based on APCV-DMV imaging features and clinical features was 22.0% greater than that of the single clinical model, and the combined model can effectively predict the early prognosis of AIS patients at 3 months, providing a strong basis for clinical decision-making.

### Funding

This work was supported by the Liaoning Provincial Joint Fund Project General Funding Scheme Project (2023-MSLH-115), Shenyang Science and Technology Bureau Public Health R&D Special Project (22-321-32-02), and Liaoning Province Applied Basic Research

Program (2023JH2/101300062). The funders had no role in study design, data collection and analysis, decision to publish, or preparation of the manuscript.

## Grant Disclosures
The following grant information was disclosed by the authors:
Liaoning Provincial Joint Fund Project General Funding Scheme Project: 2023-MSLH-115.
Shenyang Science and Technology Bureau Public Health R&D Special Project: 22-321-32-02.
Liaoning Province Applied Basic Research Program: 2023JH2/101300062.

## Competing Interests
The authors declare there are no competing interests.

## Author Contributions
- Hongyi Li conceived and designed the experiments, performed the experiments, authored or reviewed drafts of the article, and approved the final draft.
- Cancan Chang conceived and designed the experiments, authored or reviewed drafts of the article, and approved the final draft.
- Bo Zhou analyzed the data, prepared figures and/or tables, and approved the final draft.
- Yu Lan performed the experiments, analyzed the data, prepared figures and/or tables, and approved the final draft.
- Peizhuo Zang conceived and designed the experiments, analyzed the data, authored or reviewed drafts of the article, and approved the final draft.
- Shannan Chen analyzed the data, prepared figures and/or tables, and approved the final draft.
- Shouliang Qi analyzed the data, prepared figures and/or tables, and approved the final draft.
- Ronghui Ju conceived and designed the experiments, authored or reviewed drafts of the article, and approved the final draft.
- Yang Duan conceived and designed the experiments, authored or reviewed drafts of the article, and approved the final draft.

## Human Ethics
The following information was supplied relating to ethical approvals (i.e., approving body and any reference numbers):
The medical ethics committee of The People's Hospital of Liaoning Province approval to carry out the study within its facilities (approval no. (2024) H004).

## Data Availability
The radiomic features are available in the Supplemental Files.

## Supplemental Information

Supplemental information for this article can be found online at http://dx.doi.org/10.7717/peerj.19469#supplemental-information.

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
