# Peer review of "Radiomics machine learning based on asymmetrically prominent cortical and deep medullary veins combined with clinical features to predict prognosis in acute ischemic stroke: a retrospective study"

_PeerJ, doi:10.7717/peerj.19469_

## Round 0.1 · original submission · Major Revisions

The reviewers have provided comments that should help you improve your manuscript.

**Language Note:** The review process has identified that the English language must be improved. PeerJ can provide language editing services - please contact us at [email protected] for pricing (be sure to provide your manuscript number and title). Alternatively, you should make your own arrangements to improve the language quality and provide details in your response letter. – PeerJ Staff

Reviewer 1 ·

Basic reporting

All comments have been added in detail to the last section.

Experimental design

All comments have been added in detail to the last section.

Validity of the findings

All comments have been added in detail to the last section.

Additional comments

Review Report for PeerJ
(Radiomics machine learning based on asymmetrically prominent cortical and deep medullary veins combined with clinical features to predict prognosis in acute ischemic stroke: a retrospective study)

1. Within the scope of the study, various radiomics machine learning studies were carried out in patients with acute ischemic stroke based on a retrospective study.

2. In the introduction, acute ischemic stroke disease and the importance of the subject were mentioned. The literature review in this section is very limited and needs to be expanded and detailed. In addition, clearer statements and bullet points should be given at the end of the section to explain the difference of the study from the literature.

3. Approximately 5.5 years of data obtained from a hospital were used in the study. The quality of the study was increased by obtaining the dataset used in the study specific to the study. However, it was stated that the dataset distribution was made as training 70% and testing 30%. It should be clearly stated how this distribution was determined and why data augmentation steps were not needed too much.

4. It is observed that basic evaluation metric types such as Accuracy, AUC score, ROC curve and the obtained results are generally at a suitable level, albeit limited, for the study. However, obtaining missing metrics will increase the contribution of the study more.

5. It was stated that Support Vector Machine was preferred in the machine learning section. Although there are many different machine learning methods that can be used to solve this problem in the literature, please explain more clearly why SVM is preferred and/or whether different experiments have been done. The originality part in terms of the model seems limited and therefore more explanation should be made.

In conclusion, although the study attracts attention with the use of machine learning in acute ischemic stroke, attention should be paid to the sections listed above.

·

Basic reporting

The manuscript is generally well-written in professional English, though there are some minor grammatical issues that could be improved for clarity.
The literature is well referenced, covering relevant studies on stroke imaging, collateral circulation assessment methods, and applications of radiomics in AIS prediction.

Experimental design

The research question is well defined and relevant, with focus on predicting AIS patient outcomes using radiomics features of cerebral venous collateral circulation. The study fills an identified knowledge gap by applying radiomics to determine venous collateral circulation, which has been less explored than arterial circulation in AIS prognosis prediction.

Validity of the findings

The study demonstrates improved accuracy of the combined model (Clinical + Radiomics), with AUC = 0.996 in the test set. The discussion connects results to clinical relevance, emphasizing how radiomics features improve prognosis prediction.

Additional comments

This paper presents a valuable contribution to the field of AIS prognosis prediction by combining radiomics features of venous collateral circulation with clinical factors. With the suggested peer review comments - particularly regarding model comparison, feature importance analysis, and implementation considerations, this study would provide even stronger evidence for the utility of this approach in clinical practice. Thank you for this nice and well studied article.

Title:
Radiomics machine learning based on asymmetrically prominent cortical and deep medullary veins combined with clinical features to predict prognosis in acute ischemic stroke: a retrospective study


Abstract:
1. You may have to restructure Abstract into problem statement, methodology, results & significance.
2. In problem statement : include the clinical importance of prognosis prediction in Acute Ischemic Stroke and why current methods are insufficient.
3. Methodology: why radiomics and machine learning were chosen for the model development.
4. Results: How the model outperforms existing approaches in predicting prognosis. Add numerical results.


Introduction:
1. The research gap is not explicitly stated. Consider specifying limitations of existing models.
2. Include how exactly the proposed approach overcomes limitations of existing models.
3. Hard to read about novel concept as contributions are blended with background information.

Related Work:
1. Can you add subsection / information on graph-based ML models or CNN-based methods used in similar radiomics applications. Also include comparisons to recent deep learning models used in stroke imaging.
2. This section has limited discussion on how this work differs from state of the art (SOTA) models / approaches?


Methodology:
1. Mathematical formulations for feature extraction and selection ex. LASSO, SVM could have more details.
2. Need to explain equations sufficiently.
3. Are there any preprocessing steps that you followed / implemented?

Dataset:
1. Its unclear if external datasets were used for validation or if the model was trained on a single dataset. Could you add more information here?
2. Do you have more information on dataset statistics ex. number of patients, imaging modalities used, distribution of outcome classes etc.



Baseline Comparison:
1. There is not enough justification for selection of baseline models. Could you add that for readers.
2. Was there any hyperparameter tuning that was performed?

Evaluation Metrics:
1. Any justification why AUC, sensitivity, specificity, and precision were chosen as evaluation metrics.
2. Statistical significance tests ex. confidence intervals is missing.
3. This study could certainly be useful with – feature importance analysis. Is there any interpretation of why certain features improve performance.

Discussion and future work:
1. You may need to add "Limitations" subsection discussing potential biases, need for larger datasets, interpretability concerns, also challenges in clinical deployment etc.


Concussion:
2. Add a brief statement about the quantitative improvement in predictive performance achieved by the combined model.
3. Clearly mention the novelty of the radiomic approach.
4. Include more specific statement about future directions

---

## Round 0.2 · accepted · Accept

Congratulations! Your manuscript has been accepted and is ready to go to the next stage of production

Reviewer 1 ·

Basic reporting

All comments have been added in detail to the last section.

Experimental design

All comments have been added in detail to the last section.

Validity of the findings

All comments have been added in detail to the last section.

Additional comments

Review Report for PeerJ
(Radiomics machine learning based on asymmetrically prominent cortical and deep medullary veins combined with clinical features to predict prognosis in acute ischemic stroke: a retrospective study)

Thanks for the revision. The state of the paper after revision and the answers given are generally at an appropriate level. Best regards.

·

Basic reporting

This version looks good. Thanks for addressing the review comments.

Experimental design

This version looks good. Thanks for addressing the review comments.

Validity of the findings

This version looks good. Thanks for addressing the review comments.

Additional comments

This version looks good. Thanks for addressing the review comments.